# A Quantitative and Qualitative Program Evaluation of a Case-Area Targeted Intervention to Reduce Cholera in Eastern Democratic Republic of the Congo

**DOI:** 10.3390/ijerph21010027

**Published:** 2023-12-23

**Authors:** Kelly Endres, Alain Mwishingo, Elizabeth Thomas, Raissa Boroto, Wivine Ntumba Nyarukanyi, Jean-Claude Bisimwa, Presence Sanvura, Jamie Perin, Justin Bengehya, Ghislain Maheshe, Cirhuza Cikomola, Christine Marie George

**Affiliations:** 1Department of International Health, Program in Global Disease Epidemiology and Control, Johns Hopkins Bloomberg School of Public Health, Baltimore, MD 21205, USA; kendres4@jhu.edu (K.E.); liz.thomas@jhu.edu (E.T.); jperin@jhu.edu (J.P.); 2Center for Tropical Diseases & Global Health, Université Catholique de Bukavu, Bukavu B.P 265, Democratic Republic of the Congo; alainmwish@gmail.com (A.M.); raissaborotoi@gmail.com (R.B.); wivantumba@gmail.com (W.N.N.); jcbisrus@gmail.com (J.-C.B.); amani.sanvura@ucbukavu.ac.cd (P.S.); cikomola.cirhuza@ucbukavu.ac.cd (C.C.); 3Bureau de l’Information Sanitaire, Surveillance Epidémiologique et Recherche Scientifique Division Provinciale de la Santé/Sud Kivu, Ministère de la Santé Publique, Hygiène et Prévention, Bukavu B.P 1899, Democratic Republic of the Congo; justinbengehya@gmail.com; 4Faculty of Medicine, Université Catholique de Bukavu, Bukavu B.P 265, Democratic Republic of the Congo; maheshe.balemba@ucbukavu.ac.cd

**Keywords:** cholera, evaluation, emergency response, Democratic Republic of the Congo

## Abstract

Individuals living near cholera patients have an increased risk of cholera infections. Case-area targeted interventions (CATIs) promoting improved water, sanitation, and hygiene (WASH) present a promising approach to reducing cholera for those residing near cholera cases. However, there is limited evidence on the effectiveness and implementation of this approach in increasing WASH behaviors. We conducted a mixed-methods program evaluation in rural and urban eastern Democratic Republic of the Congo. The quantitative component included household structured observations and spot checks in CATI and control areas to assess WASH conditions and behaviors. The qualitative component included semi-structured interviews with CATI recipients, non-recipients, and implementers to assess CATI implementation. A total of 399 participants were enrolled in the quantitative evaluation conducted within 1 month of CATI delivery. For the qualitative evaluation, 41 semi-structured interviews were conducted, 30 with individuals in CATI areas (recipients and non-recipients) and 11 with CATI implementers. Handwashing with soap was low among both CATI and control area participants (1% vs. 2%, *p* = 0.89). Significantly more CATI area households (75%) had chlorine tablets present compared to control area households (0%) (*p* < 0.0001); however, the percentage of households with stored water free chlorine concentrations > 0.2 mg/L was low for both CATI and control area households (11% vs. 6%, *p* = 0.45). Implementers reported an insufficient supply of soap for distribution to recipients and mistrust in the community of their activities. CATI recipients demonstrated low knowledge of the correct preparation and use of chlorine for water treatment. Recipients also indicated a need for CATI implementers to engage community leaders. As CATIs are part of cholera control plans in many cholera-endemic countries, it is important to evaluate existing programs and develop evidence-based approaches to deliver CATIs that are both tailored to the local context and engage affected communities to increase WASH behaviors to reduce the spread of cholera.

## 1. Introduction

Worldwide, there are estimated to be 2.9 million cholera cases annually in cholera-endemic countries [1]. A major cholera outbreak in Africa began in 2022, impacting 14 countries including the Democratic Republic of the Congo (DRC) [2]. The DRC has one of the highest rates of cholera incidence in Sub-Saharan Africa [3,4,5,6], with over 31,000 suspected cases in the first seven months of 2023 [7]. Cholera transmission occurs primarily through inadequate water, sanitation, and hygiene (WASH) including contaminated water and poor hand hygiene practices [8,9], the impact of predictive factors such as climate change is increasingly used to guide outbreak response [10,11].

Individuals living in close proximity to cholera patients are at an increased risk of subsequent cholera infections [12,13,14]. A recent study in 2018 in eastern DRC found that individuals living within 30 m of a cholera patient were at a 20 times higher risk of developing cholera during the week after the patient’s hospitalization [15], with studies from other settings showing elevated risk continuing for one month [12,13]. To reduce cholera transmission in these hotspots, the Global Task Force on Cholera Control (GTFCC) recommends case-area targeted interventions (CATIs) where water, sanitation, and hygiene (WASH) programs are delivered within the high-risk radius around a cholera patient household. The use of CATIs for cholera control is included in the DRC National Cholera Control Plan [3,16].

CATIs are a promising rapid-response approach to reduce cholera transmission among cholera-case households and neighboring households [17,18]. After suspected cholera case identification, CATIs are deployed and generally include a combination, or “package”, of WASH interventions, including hygiene promotion, hygiene kits, active case surveillance, water source treatment, household disinfectant spraying (spraying a chlorine solution on surfaces in the household), and, less commonly, antibiotic chemoprophylaxis [19]. Some CATIs also include oral cholera vaccination (OCV), but logistical challenges have prevented widespread deployment of CATIs with OCV, and additional research is needed to determine how OCV should be delivered together with CATIs [20,21]. Hygiene promotion as part of CATIs varies and can include efforts to increase awareness of cholera transmission and prevention. Another common component of CATIs and other cholera rapid response programs often deployed during outbreak or case response and other humanitarian emergencies are “hygiene kits,” which are provided at the household level alongside hygiene promotion [22,23,24]. The exact components of hygiene kits also vary, but they generally include soap for handwashing, point-of-use (POU) water treatment (usually chlorine tablets), oral rehydration solution (ORS), a narrow-necked container for water storage, and a handwashing station [13]. CATIs have been deployed in Haiti, Yemen, Zimbabwe, Cameroon, South Sudan, Nepal, and DRC [17,19] and have shown success at promoting WASH behaviors and reducing suspected cholera cases if deployed rapidly [25,26,27]. However, given that the content and delivery of CATIs vary substantially, further research is needed on the functionality and effectiveness of different CATI packages and hygiene materials [17,25,28,29].

Even though CATIs are recommended as a cholera control strategy globally, there are a limited number of published studies evaluating the effectiveness of this approach in increasing WASH behaviors [25,26,30,31,32,33,34]. Additionally, only a few articles have utilized qualitative interviews with CATI recipients to tailor intervention design and delivery [33], and RCTs evaluating the impact of CATIs on cholera are rare [19,27]. A prospective cohort study completed in 2018 in the Kasaï-Oriental area of DRC found that delivery of hygiene kits (chlorine tablets, bar soap, and a handwashing station) and hygiene promotion to suspected cholera case households while at the health facility significantly reduced fecal contamination in stored household drinking water compared to households not receiving these kits [26]. In a process evaluation of this same study that interviewed kit recipients and implementers, the most cited barrier to kit use was the inadequate quantity of soap provided [33]. Additionally, self-reported adherence to water treatment was low and recall of water treatment methods was often incorrect [33]. Another study conducted in Haiti in 2013 that delivered hygiene kits upon discharge from the health facility found high self-reported use of the hygiene kit, with chlorine and soap as the two most popular items [25]. To build evidence on effective approaches for CATI delivery for cholera rapid response, both process and outcome evaluations are needed.

Beginning in 2020, CATIs were implemented in eastern DRC to aid in cholera response efforts. The objective of this study was to build evidence on the effectiveness of CATIs in promoting WASH behaviors and the successes and challenges of CATI delivery in the DRC. To achieve this objective, we completed structured observations in CATI recipient households and conducted interviews with CATI recipients, non-recipients, and implementers. Based on evaluation findings, we present recommendations to improve CATI delivery implemented by non-governmental organizations (NGOs) to reduce cholera in rural and urban areas in South Kivu, DRC.

## 2. Materials and Methods

For this study, we utilized quantitative and qualitative methods to evaluate the CATIs deployed between December 2022 and April 2023 in response to cholera case reports in eastern DRC. Evaluation activities were conducted across two sites in the South Kivu province of eastern DRC: Bukavu, an urban city of >1 million [35]; and Katana, a rural area along the shore of Lake Kivu. South Kivu experiences frequent cholera case reports and outbreaks, and OCV campaigns were conducted between September and October 2020 and December 2021 and April 2022 [36]. Evaluation activities included household structured observations and unannounced spot checks as well as semi-structured interviews with CATI recipients, non-recipients, and implementers.

### 2.1. CATI Delivery at Study Site

In eastern DRC at the time of this study, local NGOs partnered with UNICEF to deliver CATIs for cholera response [37]. Here, we provide a general overview of the CATI protocol used by NGOs in South Kivu. Once a suspected cholera patient is identified at the cholera treatment center (CTC), a CATI team is deployed to the patient’s household. At the household, the CATI team identifies the household’s water source and water storage practices, performs household disinfection (spraying the kitchen, toilets, floors, and door handles with a chlorine solution), and provides a hygiene kit and a short hygiene promotion message. Generally, the hygiene kit provided to households consists of 150–200 chlorine tablets for drinking water treatment, 3–5 bars of soap, 5 ORS packets, a jerry can, and informational leaflets on cholera prevention and hygiene kit component use. Hygiene promotion covers cholera transmission and prevention and how to use the items in the hygiene kit. The hygiene kit varies between settings depending on the availability of CATI materials. In some cases, a handwashing station with a lid and tap is also provided, generally given only to suspected cholera patient households.

The CATI team delivers the intervention to households within a 50–100 m radius (the high-risk area for cholera around a patient household [12,13,14]) in addition to the cholera patient household. The same hygiene kit, except the handwashing station, is given to non-cholera patient households. The CATI team commonly conducts a short door-to-door hygiene promotion session and distributes the hygiene kit to each household, followed by a group session in which all households in the specified radius participate. Delivery may vary based on the unique conditions of each CATI and household availability. The group session includes (1) information on the cause of cholera and how it can be prevented, and (2) a demonstration of the key times to wash hands with soap. WASH behaviors promoted in CATIs include water treatment with chlorine tablets, handwashing with soap, safe water storage, and food hygiene behaviors. The CATI team also identifies priority water sources for chlorination and assigns chlorinators to treat water at these points.

### 2.2. CATI and Control Area Selection

For this study, we conducted household-level structured observations and unannounced spot checks in both CATI and control areas. We defined CATI areas as villages (in rural Katana) and neighborhoods (in urban Bukavu) that received a CATI in the previous one to three weeks. Lists of areas and households that received a CATI were obtained from the local CTCs. One control area per selected CATI area was randomly selected from the same government-defined health area, based on the following eligibility criteria. In Katana, control areas were defined as villages that had not received a CATI in the previous 2 years, were more than 1 km from an area that received a CATI in the past month, and were located in the same health area as a village that received a CATI in the past month. In Bukavu, control neighborhoods were located in the same health area as a selected CATI neighborhood and were selected based on the World Health Organization Expanded Program on Immunization (EPI) sampling procedure [38]. Control area selection differed in Bukavu and Katana due to the urban, densely populated setting in Bukavu; this difference required that a village in Bukavu be a cluster of households (i.e., a neighborhood) rather than a village, like in Katana. A total of 12 CATI and 12 control areas were selected: 6 CATI and 6 control areas in Katana and 6 CATI and 6 control areas in Bukavu. Trained research officers also used the EPI sampling procedure to complete specific household selection [38]. All eligible CATI and control households had at least one adult 18 years of age or older who resided in the household 5 or more days a week. CATI households were randomly selected from the lists obtained from CTCs. Research officers selected Katana control households by standing in the center of the village and spinning a pen to select a random direction. The research officer walked in the indicated direction and counted the households from the center to the boundary of the village. A random number was then chosen to select a household along the route. From this selected starting house, the household physically nearest to the starting house was also selected for enrollment, followed by the next nearest house, and so on, until the required number of households was reached. This allowed for clustered household selection. Research officers selected Bukavu control households by choosing a random point within the relevant health area on a map, and selecting the nearest household to that point, followed by the next nearest household until the required number of households was reached. Before the start of the study, in Katana, the study team met with village chiefs to describe data collection methods and discuss any questions about the spot checks, structured observations, and semi-structured interviews that would be conducted. In Bukavu, the study team worked in coordination with local officials.

### 2.3. Part I—Quantitative Evaluation: Household Structured Observations and Spot Checks

The study team randomly selected five households from each CATI and control area for unannounced spot checks and selected another five households for structured observations. All CATI area households received a CATI between one to three weeks before the structured observation or spot check. We conducted unannounced spot checks in enrolled households to evaluate WASH conditions inside households (e.g., the presence of soap). Visits were unannounced to prevent households from preparing for our arrival. During spot checks, enumerators (trained research officers) checked for the presence of (1) cleansing agents (within 10 steps of the cooking area, within 10 steps of the latrine area, within arm’s reach of the handwashing station, and on the household compound), (2) ORS, (3) a handwashing station, and (4) chlorine tablets. Cleansing agents included bar soap, liquid soap, soapy water, and ash. Enumerators also assessed the chlorine concentration in household stored and source drinking water using a digital colorimeter (Hach, Loveland, CO, USA). Enumerators conducted spot checks immediately after household enrollment. Due to logistical challenges, spot checks were conducted in only 10 of the CATI areas (6 Katana and 4 Bukavu).

To assess handwashing with soap at key times and water treatment behaviors, enumerators conducted five-hour structured observations in households. Key times for handwashing observed were (1) before food preparation (food event), (2) before eating (food event), (3) before feeding a child (food event), (4) before serving food (food event), (5) before breastfeeding (food event), (6) after toileting (stool event), (7) after removing child feces (stool event), and (8) after cleaning the anus of a child (stool event). Structured observations were completed on the same day of enrollment after spot checks, or the next morning based on the availability of the household members and the time of day. Structured observations began between 7 AM and 12 PM. Structured observations started in the morning to capture the majority of defecation events and continued until noon to capture both breakfast and lunch food preparation events.

### 2.4. Quantitative Analysis

To compare WASH conditions and practices between CATI and control areas as assessed during spot checks and structured observations, we completed chi-square tests or Fisher’s exact tests (when 5 or fewer observations were in a category). We used logistic regression with generalized estimating equations to control household clustering to examine differences in observed handwashing behavior between CATI and control area participants. The outcome was handwashing at a food- or stool-related event at least once during structured observation and the predictor was the CATI or control area. Handwashing with soap was defined as washing both hands with a cleansing agent (bar soap, liquid soap, soapy water, or ash). We assessed both handwashing with soap and handwashing with water only.

### 2.5. Sample Size Calculation

The sample size was based on 80% power assuming that areas that did not receive a CATI would have a handwashing prevalence of approximately 5% and assuming a two-sided type one error of 0.05 for the primary statistical comparison. Given these assumptions, 24 villages having approximately 15 participants per village were needed to detect the difference between 5% handwashing and 17% handwashing with at least 80% power.

### 2.6. Part II—Qualitative Evaluation: Semi-Structured Interviews

Three research officers trained in qualitative interviewing techniques conducted semi-structured interviews with CATI implementers, recipients, and non-recipients (individuals living in households in the same village or neighborhood where the CATI was delivered, but who did not receive a CATI) to explore the successes and challenges of CATI design and implementation and recommended WASH behaviors. The same research officers conducted both qualitative interviews and quantitative observations. We asked CATI recipients about their experiences with CATI implementation, CATI materials received, and their experiences with the recommended WASH behaviors. We conducted interviews with CATI area non-recipients to better understand facilitators and barriers of WASH behaviors for those not receiving a CATI but residing in the same village or neighborhood. In interviews with implementers, we asked about their role in CATI delivery and the successes and challenges of CATI implementation. We selected implementers to interview based on their involvement with CATI delivery in South Kivu. For CATI recipients, we sampled participants across different ages, genders, and study areas (rural vs. urban).

Research officers conducted interviews in French (implementer interviews) or the Bukavu dialect of Swahili (CATI recipient, non-recipient, implementer interviews). All interviews were audio recorded. Research team members completed debriefs for all interviews to allow for iterative probing on key topics [39]. Team members used recordings of interviews and written debriefs to develop an analytic questionnaire based on emergent themes for all interviews. Two independent reviewers completed analytic questionnaires in English for all interviews and performed a thematic analysis using a deductive approach for coding themes and subthemes. The following topics from the interview guides directed code development: (1) CATI delivery, (2) handwashing with soap, (3) ORS, (4) cholera, (5) water treatment, and (6) cholera vaccine. Interviews were subsequently summarized and coded in Microsoft Word and Excel to assess key themes. Interview guides and analytic questionnaires in English are available in Appendix A. The data analysis team met weekly to discuss emergent findings. Findings on CATI successes and challenges, design, and implementation as reported by recipients and implementers were further organized according to the following implementation outcomes: adoption, acceptability, appropriateness, cost, feasibility, fidelity, penetration, and sustainability [40].

### 2.7. Ethical Approval

The protocol for this study was approved by the research ethical approval committees of the Université Catholique de Bukavu (Protocol #19728) and the Johns Hopkins Bloomberg School of Public Health (IRB00009848). Informed consent was obtained from all study participants and parental permission was obtained for participants <18 years of age.

## 3. Results

### 3.1. Part I—Quantitative Evaluation: Household Structured Observations and Spot Checks

Over the course of this study, 399 participants were enrolled in 132 total households: 209 participants in 61 CATI area households and 190 participants in 71 control area households (Table 1). Twenty-four total areas (12 villages in Katana and 12 neighborhoods in Bukavu) were included—12 CATI areas (6 Katana, 6 Bukavu) and 12 control areas (6 Katana, 6 Bukavu). A total of 58 rural households were enrolled in Katana (28 CATI, 30 control) and 74 urban households were enrolled in Bukavu (33 CATI, 41 control). Sixty-two percent (129/209) and 67% (127/190) of participants were female in CATI and control areas, respectively. CATI households were surveyed an average of 14 days after CATI area delivery (range: 5–21 days). Additional details on household and participant demographics in Katana vs. Bukavu can be found in Appendix A.

#### 3.1.1. Spot Checks

Household spot checks were conducted in 105 households (48 CATI, 57 control) (Table 2). Overall, cleansing agents were present in 35% (17/48) of CATI area and 47% (27/57) of control area households (*p* = 0.22). Presence of cleansing agents within 10 steps of the cooking area was 17% (8/48) in CATI area vs. 25% (14/57) in control area households (*p* = 0.32), and within 10 steps of the latrine cleansing agent presence was 10% (5/48) and 18% (10/57) in CATI area and control area households (*p* = 0.17), respectively. A significantly higher percentage of CATI area households had handwashing stations present compared to control households (21% (10/48) CATI area and 7% (4/57) control area, *p* = 0.047). Most CATI area households had ORS (65% (31/48)) and chlorine tablets (75% (27/36)) present, and no ORS or chlorine tablets were seen in control area households (0/57 and 0/11, respectively) (ORS *p* < 0.001, chlorine tablet *p* < 0.001). The percentage of households with stored drinking water with free chlorine greater than 0.2 mg/L was similar between CATI area (11% (4/38)) and control area households (6% (3/51), *p* = 0.45). Source water free chlorine greater than 0.2 mg/L was similar between control areas households (4% (2/48)) and CATI area households (2% (1/45) (*p* = 1.00)). Comparisons between rural and urban sites are provided in Appendix A.

#### 3.1.2. Structured Observation

Few participants were observed handwashing with a cleansing agent after a stool-related or before a food-related event, with only two participants observed handwashing with a cleansing agent at a stool- or food-related event in CATI area households (1%; 2/136) and two in control area households (2%; 2/116) (*p* = 0.89) (Table 3). Observed handwashing with water only was similar in CATI and control areas (4% (6/136) CATI vs 3% (3/116) control, *p* = 0.51). All observed handwashing with cleansing agent events were among Bukavu CATI area participants; there were no handwashing with cleansing agent events in Katana among either CATI or control area participants (Appendix A).

### 3.2. Part II—Qualitative Evaluation: Semi-Structured Interviews

#### 3.2.1. CATI Recipient and Non-Recipient Interviews

We conducted 17 interviews with CATI recipients and 13 interviews with non-recipients in CATI areas. We completed 16 interviews in Katana and 14 in Bukavu. Interview participant characteristics are listed in Table 4.

#### 3.2.2. Handwashing with Soap

Most CATI recipients correctly recalled at least one of the key times recommended for handwashing with soap. The most common correct responses were 1) before eating and 2) after using the toilet. Incorrect responses included “before doing anything you have to wash your hands” and “at any time.” Remembering to wash hands was reported as a major challenge:


*“This soap is for cleaning our hands. Often you come back from the toilet, you come back from places like the market and you forget to wash your hands and you start without remembering to wash your hands.” (CATI recipient, Bukavu, Female, 36 years of age)*


Most recipients said that they knew that the soap provided to them during a CATI was for handwashing. Despite this, many reported using the provided soap for bathing and washing clothes instead:


*“We put [soap] in the bucket for handwashing. We use it for washing our hands and even for bathing, that’s what we were told… If [others] use it for laundry, it’s because of poverty, for lack of any other alternative.” (CATI recipient, Katana, Female, 36 years of age)*


Despite understanding the importance of using soap to prevent cholera, the lack of funds for additional soap was highlighted by many recipients:


*“But how can we manage to stay clean when we don’t have the financial means? Cleanliness requires financial resources.” (CATI recipient, Bukavu, Male, 35 years of age)*


Almost all recipients and non-recipients noted that they knew to use ash for handwashing when there was no soap available, and most said that they had used ash at some point, or intended to use ash when their soap ran out. A few interview participants mentioned that ash should also be placed in the latrine—one recipient reported learning this during a CATI in Bukavu, and one non-recipient said they had been taught this during previous cholera awareness interventions conducted in 2018 in the Katana area.

#### 3.2.3. Recommendations for CATI Design on Handwashing with Soap

To reduce the likelihood of households using bar soap for purposes other than handwashing, we recommend the distribution of premade soapy water made using water and detergent powder in a recycled plastic bottle and encouraging households to make their own soapy water with water and small leftover pieces of bar soap. Behavior change communication should emphasize that the provided soap or other cleansing agent should be used for handwashing during the high-risk period for cholera. Hygiene promotion programs should be developed using theory-driven and evidence-based approaches [41]. Placement of ash in a container alongside water at latrines and cooking areas could be promoted to facilitate handwashing with a cleansing agent if soap is not available.

#### 3.2.4. Water Treatment

Most households in both Bukavu and Katana reported that they did not treat their drinking water before CATI delivery (for non-recipients, this represents their current water treatment). Recipients and non-recipients used a variety of water sources such as lake water, stream water, or tap water. Rainwater was mentioned as a potential source of water that could be treated with chlorine tablets to reduce cholera:


*“They gave us materials such as [chlorine] tablets to put in the water so that when we want to drink water, we can even use rainwater in which we put the tablet to avoid catching cholera.” (CATI recipient, Bukavu, Male, 20 years of age)*


Almost all CATI recipients mentioned using chlorine-treated water for handwashing, bathing, cooking, and other household chores like laundry, with handwashing and bathing reported most often. One recipient noted his appreciation for the chlorinated water for bathing because it cured his family’s skin rashes:


*“[Chlorine-treated water is] good for bathing, but also for washing dishes, drinking, cooking, washing clothes, but much more so for washing food... [We like the chlorine tablets] for cleanliness, because some of my children used to suffer from dermatitis, and all this was due to dirt.” (CATI recipient, Bukavu, Male, 35 years of age)*


While chlorine tablets were reportedly very easy to use, most CATI recipients could not explain how to correctly treat drinking water using chlorine tablets (the correct treatment for tablets provided was 4 tablets in 20 L of water and waiting 30 min before consumption):


*“Well, I know that it’s water purification tablets, you put a tablet in 20 L and wait a few minutes before using it.” (CATI recipient, Bukavu, Male, 35 years of age)*


In some instances, recipients reported that chlorine tablets of varying doses were provided in the same area:


*“There are those who use 4 tablets in twenty liters of water and others 1 tablet … All these tablets are the same because they all taste the same... After fetching the water, we put in one chlorine tablet and wait thirty minutes. If it’s the other type of chlorine tablet, which requires 4 tablets, we wait 10 min, as we were instructed, and then we can start drinking the water.” (CATI recipient, Katana, Female, 36 years of age)*


Many recipients and non-recipients also said that they did not like the taste and smell of chlorinated water for drinking and disliked the taste of the food prepared with chlorinated water.

One challenge that was often mentioned was that chlorine tablets needed to be replenished. There was little difference in reports of this challenge between CATI recipients and non-recipients. Some recipients additionally felt that the CATI had not provided enough tablets:


*“First of all, we need you to provide us with a sufficient number of chlorine tablets, because giving us a single tablet does not guarantee what will happen next, especially as we have to wait for you to come.” (CATI recipient, Bukavu, Male, 35 years of age)*


Both recipients and non-recipients in Katana reported other challenges, including the dispensary and health centers not having chlorine tablets in stock. Some recipients, both in rural and urban areas, reported not treating water after the chlorine tablets were finished, often due to the cost of tablets. Some CATI recipients relayed that they discussed the chlorine tablets they received with their neighbors, who they said often did not know how to use tablets or how to access them. One recipient noted that their neighbors reprimanded them for using treated water because they would lose their tolerance for drinking untreated water:


*“All I see are some neighbors who refuse to drink the water, saying that it doesn’t smell right. I tell them that I’m already used to it, and they tell me that when the stock of medicines runs out, we’ll fall ill because we’ll be used to it even though we don’t have the means, and when we stop, we’ll catch cholera.” (CATI recipient, Bukavu, Female, 33 years of age)*


A few recipients said that they shared chlorine tablets with their neighbors. One described how his neighbors asked for tablets when they became sick, after seeing that the households who received tablets were not getting sick:


*“Interviewer: Did your neighbors use the chlorine tablets you gave them? Why?”*



*“Respondent: Yes, they saw what we had cholera, and they had noticed that when we used these tablets, we didn’t get sick anymore...Those who live close to me treat their drinking water with the tablets we gave them, but others wait until they show signs of illness to come and ask us for tablets.” (CATI recipient, Katana, Male, 45 years of age)*


Some recipients and non-recipients said that their neighbors thought that chlorine tablets were only needed when someone in the household was ill:


*“Interviewer: What did the neighbors think about the use of chlorine tablets?”*



*“They just left it, without using them...They don’t really use it, they just keep it.”*



*“Interviewer: Why?”*



*“They have the impression that it’s only those who were sick who are concerned.” (CATI recipient, Katana, Female, 36 years of age)*


#### 3.2.5. Recommendations for CATI Design on Water Treatment

Implementers delivering CATIs should clarify that chlorine-treated water is only needed for drinking water (not bathing) and emphasize that using chlorine-treated water for other purposes will mean running out of tablets before the end of the high-risk period for cholera transmission. CATIs could also include a group session where recipients drink chlorine-treated water together and discuss taste and smell [42]. Implementers should provide clear and consistent instructions on how to treat and when to treat water with chlorine tablets and ensure leaflets with pictorial instructions are given. Only one type of tablet should be delivered to avoid confusion. Finally, CATIs should recommend where households can purchase chlorine tablets, verify availability at recommended locations, or recommend boiling as an alternative method of drinking water treatment once supplies are exhausted.

#### 3.2.6. ORS

Most recipients were well-informed about the use of ORS, reporting that it is used for diarrhea and cholera. Most also accurately described ORS preparation. They further mentioned that if diarrhea persists after using the ORS solution, the patient should be taken to the hospital for additional treatment.


*“Well, they told us that if a child has diarrhea more than once [in a short time], you can take [ORS], put it in the cup and give it to the child… if it persists, we have to take the patient to the hospital... This serum helps us in case of dehydration, because in this situation the oral serum is used to compensate the water loss, before going to the hospital, we should have already consumed the oral serum.” (CATI recipient, Bukavu, Male, 35 years of age)*


However, one recipient living in Katana thought that ORS functioned similarly to chlorine tablets by preventing disease:


*“What do you know about [ORS]?”*



*“This serum is also interesting because if someone consumes it, they can prevent sickness.” (CATI recipient, Katana, Male, 45 years of age)*


#### 3.2.7. Recommendations for CATI Design on ORS Use

We recommend that during CATI delivery, the purpose of ORS be clearly explained: that it is for rehydration during severe diarrhea but does not remove germs from water or prevent disease.

#### 3.2.8. CATI Materials and Delivery

Some recipients reported receiving CATIs delivered in a door-to-door format only, while others reported group sessions or a combination of door-to-door and group sessions, with similar variation in rural and urban areas. Most recipients stated that they preferred group sessions, saying that if they forgot information or missed part of the CATI, they could ask their neighbors. Many recipients reported not receiving all intervention materials that, according to the CATI procedure, should have been provided. Almost all households received ORS and chlorine tablets. However, most interviewed recipients in Katana said they had not received soap, while most recipients in Bukavu reported that they had. Recipients felt that the quantity of soap given was insufficient, pointing out that when supplies run out, cholera may reappear. Most recipients who received household disinfection expressed appreciation for it, saying that it helped prevent cholera in their household. One recipient in Bukavu felt that household spraying was important because “if that bad air is in the house, it’s time to get rid of it.”

Recipients said they generally liked CATI and felt that increased awareness would help reduce the risk of disease in their communities.


*“My own wife had been affected by cholera, so they decided to send a group of people to visit us, and if they hadn’t come, everyone else would have been contaminated. God helping, she was well taken care of and the teachings were well respected, which spared the contamination of other family members.” (CATI recipient, Bukavu, Male, 35 years of age)*


When asked about recommendations for future interventions against cholera, recipients had several ideas. Many recommended additional campaigns to raise awareness for others who did not yet have information about cholera, or for those who were not practicing CATI-targeted behaviors fully or correctly. A few recipients and non-recipients also recommended that CATIs engage community leaders such as community chiefs and health workers to help encourage the adoption of protective behaviors.


*“Let’s all help each other to stop the uncleanliness. We can even ask the neighborhood chief to help us eradicate this uncleanliness.” (CATI recipient, Bukavu, Male, 14 years of age)*


CATI recipients often recommended increasing the quantity of items in the hygiene kit, including chlorine tablets and soap. Others mentioned a need for improved water sources with methods for continued maintenance and upkeep:


*“Make this [source] water your priority, look at the priority areas and plan the installation of water sources that we will maintain well.” (CATI recipient, Bukavu, Male, 35 years of age)*


#### 3.2.9. Recommendations on CATI Design for Intervention Delivery

CATI implementers should ensure complete and consistent delivery of all intervention materials, including soap and chlorine tablets. Standardized protocols are needed for the delivery of CATIs. Finally, all CATIs should be implemented in partnership with local stakeholders such as community chiefs or local officials to facilitate acceptability.

#### 3.2.10. Vaccination

Recipients said that information about the cholera vaccine was not provided during CATI delivery. Most recipients and non-recipients in both Bukavu and Katana had not heard of the cholera vaccine, and only a few individuals said that they had received cholera vaccine, all of whom resided in Katana and were non-recipients (Table 4). About half of the recipients and non-recipients who had heard of the cholera vaccine asserted that it prevented any future cholera infections and that if a vaccinated individual drank cholera-contaminated water, they would not become ill. They expressed surprise to see vaccinated individuals falling ill.


*“How can we still contract this disease of diarrhea and vomiting when we’ve taken this vaccine? The children were vaccinated and even us parents.” (Non-recipient, Katana, Female, 37 years of age)*



*“Interviewer: After a person receives a cholera vaccine can they still get cholera?”*



*“No... You can’t get sick again because you’ve already taken the treatment [vaccine].” (CATI recipient, Bukavu, Female, 36 years of age)*


One non-recipient expressed concern that they received an expired or low-quality vaccine because a vaccinated family member still died from cholera:


*“This vaccine was taken, but until now cholera persists, I had taken it and even my family members had taken it, but it didn’t prevent my father from dying even though he had taken the vaccine... I don’t know if the vaccine was damaged or expired, I don’t know because there are people who took it but got sick after.” (Non-recipient, Katana, Male, 43 years of age)*


Another non-recipient noted that the vaccine was effective unless you are under the effects of a curse.


*“Once you’ve been vaccinated against cholera, you can rarely catch it. Unless you’re under a curse, you can’t catch it.” (Non-recipient, Katana, Female, 27 years of age)*


#### 3.2.11. Recommendations on CATI Design for Cholera Vaccination

CATI teams should mention any ongoing cholera vaccine campaigns in the areas where CATIs are conducted and provide information on what the cholera vaccine is and its effectiveness. Teams should explain that the cholera vaccine reduces the risk of cholera but does not eliminate the risk, which is why preventative behaviors remain important—vaccinated individuals still need to drink treated water and wash their hands with soap to reduce the risk of cholera.

#### 3.2.12. Implementer Interviews

We interviewed 11 CATI implementers involved in South Kivu, DRC cholera response efforts for this study. Implementers included South Kivu provincial and health zone supervisors and staff, cholera prevention and rapid response team supervisors and promoters, and medical staff at the CTC in Bukavu and local health facilities.

#### 3.2.13. CATI Delivery

Implementers reported a variety of challenges related to CATI delivery, mainly focused on the lack of financial, physical, and human resources for CATI delivery:


*“There was also a shortage of inputs, including chlorine and the aquatabs that were used. These were often the big challenges.” (Implementer—Bukavu)*


Other challenges discussed by implementers included accessing households for CATI delivery due to weather and the remote location of some rural households:


*“It’s just that when it rains with the land here, it’s complicated to reach the case, secondly there are the cases that are isolated, and to reach them you have to climb the mountains because the vehicles can’t get there.” (Implementer—Katana)*


Additionally, the CATI rapid response team supervisors that we interviewed discussed that some households refused CATI delivery altogether, or refused certain aspects of it, such as household spraying or hygiene kits. One implementer expressed mistrust in the community of their activities:


*“There are always challenges, every day there are challenges. There are times when we meet a community that is stubborn and refuses, for example, that the disinfectors can’t come and spray in their homes and they don’t believe in cholera and there are people who say: ‘The products you’re bringing in are going to make us sick, we don’t want you to bring that into my home.’ There are those who refuse to have their houses disinfected, there are those who refuse to open their doors, there are those who refuse to take preventative products, for example water treatment products, or products like ORS, they say ‘No, no, these are things that come from white people, we don’t want these things, we don’t want our houses sprayed, thank you.’” (Implementer—South Kivu)*


Implementers also described challenges with encouraging behavior change in recipient communities.


*“We were working on the preventative side, where we were working on people’s mentalities, and changing mentalities takes a lot of time, it’s a long process. It’s still not easy. You can talk to people, but even if they’re informed, even if they’re aware, even if they’re educated, they still express opinions that are always contrary to the messages we give them.” (Implementer—South Kivu)*


Despite these challenges, implementers felt that CATIs were successful in reducing cholera cases and raising hygiene awareness.

Implementers gave several recommendations to improve CATIs for future deployments. The first recommendation was to reduce reliance on external partners for prevention programs, as this was seen to reduce flexibility by delaying the re-activation of CATIs between cholera case clusters and outbreaks and limiting access to areas not covered by allotted funding from external partners. Second, implementers also recommended maintaining prevention activities even when there are not active cholera outbreaks or cases, to reinforce the importance of drinking water treatment and handwashing with soap over time. Third, implementers also wanted to prioritize increased community engagement through the involvement of community leaders from the beginning of program implementation, both to decrease the bureaucracy involved during the time of cholera response (time spent meeting with local leaders before CATI delivery during the time of outbreaks) and to improve hygiene promotion activities both within and outside the time of cholera outbreaks. Finally, almost all implementers mentioned the lack of access to improved water sources in the cholera-impacted communities they were targeting, and the need to improve water sources in parallel to CATI delivery. One implementer specifically criticized the CATI approach for not including infrastructure improvements of water sources:


*“With the approach that we have, the approach that is downright urgent and does not have the basis of a sustainable development project, we are trying to carry out activities that seek to cut the chain of transmission during the epidemic. This is not, from my point of view, a sustainable solution to the problem of cholera in South Kivu. When it comes to drinking water sources in South Kivu, I have already visited all the areas where there is cholera, and I have noticed that the problem is linked to the lack of drinking water, and everywhere we have found that with the CATI project, the CATI approach does not really have a line for, for example, the rehabilitation of springs. This means that we can activate chlorination sites that are mobile and dynamic, taking into account the dynamics of the epidemic, but this is not really a solution for these active areas with cholera cases. This is really a shortcoming; this is really the challenge–when there’s a project that fits in with the emergencies and doesn’t provide the real solution to the problem.” (Implementer—South Kivu)*


#### 3.2.14. Recommendations on CATI Design from Implementers

CATI implementers should prioritize increased engagement with community leaders and other local stakeholders for CATI program delivery to build trust and acceptance of the promoted behaviors in the community, and to facilitate rapid response. Additionally, sufficient resources, particularly of WASH materials, need to be allocated for CATI delivery. CATI programs should also be integrated with infrastructure programs to increase access to improved water sources within 30 min of CATI households (basic improved water sources).

#### 3.2.15. Overall Perceptions of Implementation Outcomes

We assessed CATI delivery using eight implementation outcomes: acceptability, adoption, appropriateness, cost, feasibility, fidelity, penetration, and sustainability [40]. Table 5 reviews CATI recipient and implementer perceptions on CATI delivery for these 8 outcomes.

Overall, CATI recipients and implementers generally perceived the program to be acceptable and appropriate but also noted key challenges. Most recipients agreed that the CATI program aimed to prevent cholera in the household, and they appreciated individuals coming to their communities to deliver CATIs. However, implementers described low acceptability of CATIs in some areas, including instances where community members refused entry of CATI staff into their homes for program delivery.

The fidelity of CATIs, assessed through interviews with recipients and implementers, was low and varied substantially between areas where CATIs were delivered. Both recipients and implementers reported insufficient quantities of WASH materials for program delivery as a major challenge. Furthermore, the descriptions of CATI delivery reported by recipients varied widely, with some recipients reporting door-to-door delivery, some group delivery, and some both delivery types. Penetration, assessed only at the implementer level, was generally high, though implementers described some refusal of CATI household visits for spraying and CATI materials among target households.

Cost was a challenge mentioned by both implementers and recipients. Implementers discussed the challenges with funding the overall CATI program and providing consistent supplies of hygiene kits and other materials. Recipients noted that replenishing stocks of soap and chlorine tablets was a financial hardship. The feasibility of CATI delivery was challenged at the implementer level by the mountainous terrain and rainy conditions as well as by procurement and supply of WASH materials. At the recipient level, feasibility challenges were not directly explored. Recipients and implementers shared mixed views on the sustainability of CATIs, with limited ability to continue WASH behaviors long-term and limited provision of improved safe-water sources mentioned. Table 6 summarizes recommendations made to improve CATI delivery and WASH behavior uptake based on study findings.

## 4. Discussion

In this combined quantitative and qualitative program evaluation we assessed the uptake of recommended WASH behaviors after CATI program delivery and identified successes and challenges of CATI program design and implementation in South Kivu, DRC. We identified low uptake of the targeted WASH behaviors through structured observations and spot checks of both CATI and control households. Through qualitative interviews, we explored successes and challenges with both CATI delivery and uptake of targeted WASH behaviors. From these findings, we have made key recommendations on how to improve CATI delivery in this setting.

Handwashing with soap is a crucial intervention to break diarrheal disease transmission [43]. However, handwashing with soap was very low both among CATI recipients and in control households in both urban and rural areas in South Kivu, DRC. This result aligns with our qualitative findings, where we found that few CATI recipients in Katana said that they had received soap in their hygiene kit. From interviews, we also learned that recipients had used soap from hygiene kits for household tasks, such as bathing and washing clothes, rather than for handwashing. Our findings are similar to a 2019 study in North Kivu, DRC, which found that failure to provide hygiene materials to households was a major barrier to the successful promotion of handwashing with soap in internally displaced persons camps [44]. Given the challenges with the provision of bar soap, alternative options such as promoting the use of small pieces of leftover soap and water in a recycled bottle to prepare soapy water or the use of ash, could be promoted, as was done in our recently developed Preventative Intervention for Cholera for 7 Days (PICHA7) WASH program for cholera patient households in South Kivu, DRC [45]. Distribution of premade soapy water using water and detergent powder in a recycled plastic bottle may also be a promising alternative to bar soap, since it may be less likely to be used for purposes other than handwashing. Additionally, hygiene promotion for handwashing during CATIs should incorporate theory-driven and evidence-based behavior change communication and engage the community in intervention development to encourage the uptake of targeted WASH behaviors [41]. Behavior change communication programs should target behavioral determinants associated with the targeted outcome behaviors using behavior change frameworks such as the Integrated Behavioural Model for WASH [41]. In eastern DRC, perceived susceptibility to disease and self-efficacy were associated with higher handwashing with soap [46] and could be targeted in CATI hygiene promotion. In addition, it is possible to send hygiene promotion mobile messages to CATI households as behavior reminders as is done for the PICHA7 WASH program in South Kivu, DRC [45]. Future CATI implementation should prioritize the provision of a cleansing agent that can be provided to all program households, and tailor hygiene promotion to the target population through community engagement and behavior change theory.

Significantly more CATI area households (75%) had chlorine tablets present compared to control area households (0%). However, the percentage of stored water with chlorine concentrations >0.2 mg/L was low for both CATI and control area households (11% vs. 6%). This finding suggests that while CATI area households had more chlorine tablets compared to households that did not receive CATI, they were likely not using them at the time of this visit, or not using them correctly, to treat their drinking water. Consistent with this explanation, in interviews, CATI recipients mentioned several reasons for not using the tablets, such as a neighbor’s belief that using the tablets would make them more susceptible to disease when drinking untreated water later, or that the tablets should only be used during periods of active illness. In addition, there was low recall of the correct dosing and treatment time for chlorine in drinking water, chlorine-treated water was used for bathing and other household tasks, and one participant expressed the belief that ORS could also be used as a water treatment method. Low recall and the use of chlorinated water for a variety of functions were similar for CATI recipient and non-recipient participants. The use of chlorinated water for handwashing and bathing may stem from Ebola outbreaks in eastern DRC, where this was recommended, or from experience from visiting CTCs in South Kivu that often provide chlorinated water for handwashing for patients and visitors. Incorrect recall of drinking water treatment methods and low self-reported adherence to POU water treatment were also noted in a previous DRC study that distributed hygiene kits to suspected cholera patient households in health facilities [33]. Limited recall of treatment methods may contribute to decreased use of POU water treatment [47,48]. We recommend that CATI delivery include a demonstration of the correct use of chlorine tablets and a pictorial cue card explaining how to treat water using chlorine tablets. We also suggest that chlorine tablets with the same dosing be delivered within the same community during CATI delivery and over time, to avoid confusion about dosing instructions.

Many participants that we interviewed expressed concerns about the taste of chlorinated water. To address this concern, we recommend CATI delivery include a group session where recipients drink chlorine-treated water together and have a discussion about the taste. This method was previously used in both Bangladesh and DRC at the patient’s bedside to increase the acceptability of chlorine tablets [42,45]. Given that interview participants said that procuring chlorine once the CATI-provided amount was exhausted was both a financial and logistical issue, we also recommend that CATI implementers provide information about where chlorine tablets can be procured, recommend boiling as an alternative drinking water treatment method, and discuss the alternative financial burden to the household if someone becomes ill with cholera (e.g., time away from work due to illness or taking care of love ones) as motivation to sustain water treatment. CATI implementers should also prioritize the development of more improved water sources and ensuring safely managed drinking water including prioritizing chlorination and water quality monitoring of existing piped water networks, as requested in interviews by CATI recipients and almost all implementers. 

Household spot checks and CATI recipient and implementer interviews indicated major challenges with the delivery of WASH intervention materials. Most studies of CATIs and other cholera rapid response delivery programs have reported a variety of challenges with the complete delivery of WASH materials, including logistical and financial constraints [30,33,49]. Incomplete delivery of WASH materials has the potential to both decrease the effectiveness of a CATI and to decrease trust between recipients and CATI teams, as has been reported previously [30]. Before cholera outbreaks, funding, implementation, and Ministry of Health partners should work together to source adequate funding and supplies.

There was limited knowledge of cholera vaccine in our study areas in South Kivu, DRC, with only a few recipients reporting that they had received cholera vaccine and only at our rural site. Furthermore, for those who were vaccinated, there was a surprise that the vaccine had not provided complete protection against cholera for themselves and those in their household and community. Numerous studies have described the potential benefits of cholera vaccine delivery inclusion in CATIs [12,17,21,50], though studies of real-world applications are limited and ongoing, often hampered by limited vaccine availability due to the low cholera vaccine stockpile globally [20,21,31,51,52]. In addition, with limitations in access to OCV supplies, challenges with OCV include the need for multiple doses, incomplete protection provided by vaccination, and low community awareness [53]. Although we did not explore vaccine hesitancy, a recent CATI with OCV intervention study in Cameroon briefly described this as a potential challenge [20]. While CATIs are ultimately delivered to small groups of people in hotspots for cholera, increased awareness of OCV could help ease larger cholera vaccination campaign rollout when these occur. However, additional research is needed to understand if skepticism of the cholera vaccine would decrease credibility for other CATI promotion activities. We recommend that if OCV is not included in the intervention package, CATIs should at least include information on ongoing cholera vaccination campaigns in nearby areas and information on OCV’s purpose and effectiveness.

From interviews, we also learned that one CATI implementer encountered mistrust and hesitancy toward CATI activities in target areas, particularly at rural sites. This has been reported previously in CATI delivery in Nigeria [30]. Notably, while CATI implementers stated that they experienced distrust in communities regarding household disinfection, none of our CATI recipients discussed this. Additional interviews are needed to understand this difference. Our finding of distrust and hesitancy indicates the need to further engage with local and community stakeholders to improve community acceptance, which was recommended by both CATI recipients and implementers in our qualitative evaluation.

Finally, there was a lack of standardization for how CATIs were delivered in the same areas, with the mode of delivery varying widely between door-to-door, only group, or both sessions. While acknowledging some need for flexibility, we recommend that implementers prioritize delivering both group and door-to-door sessions. Including both session types is beneficial as group sessions may facilitate discussions among community members, while door-to-door delivery allows for individualized intervention delivery tailored to the needs of individual households, as was found in our recent study in Bangladesh [54]. Furthermore, a more standardized delivery method with standardized CATI protocols ensures that materials and hygiene promotion are delivered together and reduces the risk of key components of intervention delivery being missed.

The findings from this study emphasize the importance of combined quantitative and qualitative program evaluations of WASH interventions and the need for evidence-informed approaches to improving CATI delivery. At our study site in DRC, our research group recently developed the PICHA7 WASH program, which promotes handwashing with soap, water treatment, and safe water storage to cholera patient households during the 7-day high-risk period for cholera infections after the presentation of the index patient in their household to the health facility for treatment [45]. The PICHA7 program was developed through 18 months of formative research to tailor this program to our partner communities and took a theory-driven approach informed by IBM-WASH [41]. Through this formative research, which engaged community members and programmatic stakeholders, we were able to learn about the key barriers and facilitators to the promoted WASH behaviors and target these in PICHA7 intervention development [45]. We then piloted the program and obtained beneficiary feedback, which was used for further program refinement. In our RCT of the PICHA7 WASH Program (NCT05166850) in Bukavu, South Kivu, DRC we found that delivery of PICHA7 significantly increased handwashing with soap, and stored drinking water with free chlorine >0.5 mg and significantly reduced hospital admissions for diarrhea and rice water stool (a hallmark sign of cholera) at our 12-month follow-up in the PICHA7 compared to the standard recommendation arm, which only received a basic WASH recommendation and instructions on the correct use of ORS (George et al. 2024 submitted). These findings demonstrate that a theory-informed, evidence-based intervention approach that engages communities and is tailored to the intervention setting can facilitate WASH behavior change.

This study has limitations. First, all interviews were conducted in South Kivu, which limits the transferability of findings to other areas where CATI programs are implemented. Future similar studies are needed in other cholera-endemic areas of DRC and globally. Second, we only evaluated the uptake of WASH behaviors in a 1-to-3-week period after CATI program delivery. However, given the low WASH uptake observed during this period, it is unlikely that WASH behavior increased further over time among CATI households.

This study also has several strengths. First, the use of both quantitative and qualitative methods to assess CATI program delivery. This approach allowed us to combine assessments of WASH behavior and availability of WASH materials with feedback from CATI recipients and implementers, providing robust insights into facilitators and barriers to effective CATI delivery in our study setting. Second, the inclusion of control areas in our quantitative evaluation to assess WASH behaviors and materials in households that had not received a recent CATI. This provided insights into the differences in WASH behaviors between similar households that had received a CATI, and those that had not. Finally, we conducted activities in both a rural and urban context, expanding the transferability of our results.

## 5. Conclusions

Our quantitative and qualitative program evaluation of CATI delivery in DRC identified important challenges to program implementation to target in future CATI programs in this setting. Findings included low uptake of WASH behaviors, low acceptability of CATIs in some areas and mistrust of CATI implementers, lack of community engagement in program delivery, incomplete delivery of WASH materials, lack of a standardized approach for program delivery, incorrect or incomplete recall of how to perform WASH behaviors, and challenges with program sustainability. Through this program evaluation, we developed recommendations on CATI program delivery to improve future program implementation. The findings from this evaluation emphasize the importance of engaging communities in CATI program development to ensure effective program implementation to increase WASH behaviors and reduce cholera in DRC.

## Figures and Tables

**Table 1 ijerph-21-00027-t001:** Participant and household demographics by CATI delivery and control areas in South Kivu, DRC.

	CATI Area			Control Area			*p*-Value
	%	n	N	%	n	N	
Participants			209			190	
Participants per household	3.4 ± 3.2 (1–13)			2.7 ± 2.6 (1–9)			
Households			61			71	
Female	62%	129	209	67%	127	190	0.29
Age (years, all)			209			190	0.68
0–2	10%	21		9%	17		
2–5	12%	24		12%	22		
5–12	21%	44		17%	32		
12–18	57%	120		63%	119		
Villages/neighborhoods			12			12	
Households per village/neighborhood	5.1 ± 2.7 (1–11)			5.9 ± 2.3 (2–12)			
Water Source Type ^§^			48			57	0.96
Public tap/standpipe	44%	21		47%	27		
Protected dug well	27%	13		23%	13		
Unprotected dug well	19%	9		18%	10		
Tube well	6%	3		5%	3		
Other	4%	2		7%	4		
Days since CATI delivery *	13.8 ± 4.2 (5–21)						

SD = standard deviation. n indicates the number of participants in the category. N indicates the total number of participants. Households per village/neighborhood, participants per household, and days since CATI delivery were reported as Mean ± SD (min–max). *p*-values calculated with the chi-square method. * Time since CATI delivery calculated at the household level. ^§^
*p*-value calculated with Fisher’s exact test. CATI = case-area targeted intervention.

**Table 2 ijerph-21-00027-t002:** Spot checks of WASH indicators comparing CATI delivery and control areas in South Kivu, DRC.

	CATI Area			Control Area			*p*-Value
	%	n	N	%	n	N	
Households			48			57	
Villages/neighborhoods			10			12	
Households per village/neighborhood							
Mean ± SD (min–max)	4.8 ± 2.0 (3–9)			4.8 ± 1.6 (2–8)			
The cleansing agent present in the home	35%	17	48	47%	27	57	0.22
Cleansing agent present within 10 steps of the latrine	10%	5	48	18%	10	57	0.17
Bar soap *	8%	4	48	18%	10	57	0.25
Liquid soap	0%	0	48	0%	0	57	-
Soapy water	0%	0	48	0%	0	57	-
Ash	0%	0	48	0%	0	57	-
Water and cleansing agent *	2%	1	48	9%	5	56	0.21
Cleansing agent present within 10 steps of the cooking area	17%	8	48	25%	14	57	0.32
Bar soap	17%	8	48	21%	12	57	0.57
Liquid soap *	0%	0	48	2%	1	57	1.00
Soapy water *	0%	0	48	2%	1	57	1.00
Ash	0%	0	48	0%	0	57	-
Water and cleansing agent	10%	5	48	21%	12	57	0.14
Handwashing station present *	**21%**	**10**	**48**	**7%**	**4**	**57**	**0.05**
Cleansing agent within arm’s reach of handwashing station *	8%	4	48	2%	1	57	0.18
Bar soap *	8%	4	48	2%	1	57	0.18
Liquid soap	0%	0	48	0%	0	57	-
Soapy water	0%	0	48	0%	0	57	-
Ash	0%	0	48	0%	0	57	-
ORS present in the home *	**65%**	**31**	**48**	**0%**	**0**	**57**	**<0.0001**
Chlorine tablets present in the home *	**75%**	**27**	**36**	**0%**	**0**	**11**	**<0.0001**
Stored water-free chlorine > 0.2 mg/L *	11%	4	38	6%	3	51	0.45
Stored water total chlorine > 0.2 mg/L	16%	6	38	12%	6	51	0.58
Source water-free chlorine > 0.2 mg/L *	2%	1	45	4%	2	48	1.00
Source water total chlorine > 0.2 mg/L *	2%	1	45	15%	7	48	0.06

SD = standard deviation. One spot check is conducted per household. Cleansing agents include bar soap, liquid soap, soapy water, or ash. * *p*-value calculated with Fisher’s exact test. No *p*-value is calculated if no observations of a specific category are recorded. Significant findings are in bold. CATI = case area targeted intervention.

**Table 3 ijerph-21-00027-t003:** Logistic regression analysis of participants handwashing with a cleansing agent during 5-h structured observation by CATI delivery area vs. control (predictor) during household structured observation in South Kivu, DRC.

	CATI Area			Control Area			Logistic Regression	
	%	n	N	%	n	N	OR (95% CI)	*p*-Value
Participants handwashing with a cleansing agent at stool and food events								
All events	1%	2	136	2%	2	116	0.88 (0.13, 5.83)	0.89
Stool events *	2%	1	53	0%	0	53	-	0.49
Food events	1%	1	126	2%	2	107	0.44 (0.043, 4.49)	0.49
Participants handwashing with water only at stool and food events								
All events	4%	6	136	3%	3	116	1.63 (0.38, 7.00)	0.51
Stool events	0%	0	53	0%	0	53	-	-
Food events	5%	6	126	3%	3	107	1.67 (0.39, 7.18)	0.49

OR = odds ratio. CI = confidence interval. Food events include food preparation, before eating, before feeding someone, before serving food, and before breastfeeding. Stool events include after toileting, after removing child feces, and after cleaning the anus of a child. Cleansing agents include bar soap, liquid soap, soapy water, and ash. Only bar soap and liquid soap were observed to be used by participants for handwashing. * *p*-value calculated with Fisher’s exact test by household level. No OR was calculated due to the small sample size. CATI = case-area targeted intervention.

**Table 4 ijerph-21-00027-t004:** Demographics of semi-structured interview participants.

	CATI Recipients and Non-Recipients		Implementers	
	%	n	%	n
Interviews		30		11
Gender				
Male	50%	15	100%	11
Female	50%	15	0%	0
Age (years)				
12–18	30%	9	0%	0
Over 18	70%	21	100%	11
Study Area				
Bukavu	47%	14	-	-
Katana	53%	16	-	-
Cholera Vaccine				
Heard of the cholera vaccine	37%	11	-	-
Received vaccine	17%	5	-	-
Knows others who received the vaccine	23%	7	-	-

CATI = case-area targeted intervention. Recipients lived in households that received a CATI. Non-recipients were individuals living in households that did not receive a CATI, but resided in the same village or neighborhood where the CATI was delivered, i.e., lived in a CATI area.

**Table 5 ijerph-21-00027-t005:** Implementer and Recipient perceptions of CATI delivery.

Outcome	Recipient Perceptions	Implementer Perceptions
Acceptability	Most recipients appreciated CATI delivery to help prevent cholera in their households. However, some reported disappointment with the quantity of materials delivered, and said they would prefer larger quantities of consumable products, such as soap and chlorine tablets. Others reported incomplete delivery of the intervention package (e.g., no receipt of soap).	Implementers noted refusal to allow household disinfection, refusal of hygiene kit delivery (soap, chlorine tablets, and oral rehydration solution), mistrust of CATI materials, and disbelief in cholera among CATI recipients. However, implementers reported that communities generally accepted CATIs. Implementers believed that CATIs successfully aided in decreasing the spread of cholera outbreaks.
Adoption	Adoption of targeted WASH behaviors was low. Recipients reported using materials for other purposes, such as using chlorinated water for handwashing and soap for laundry.	Implementers believed that CATIs successfully increased WASH behaviors in communities where CATIs were delivered, however also stated that behavior change was challenging in some communities and takes time.
Appropriateness	Recipients generally felt that CATIs were beneficial for their households. Some recipients believed improving local water sources would be more impactful.	Implementers stated that CATIs prevented cholera outbreaks. Water source improvements were presented as an important method to reduce future outbreaks.
Cost	The cost of soap and chlorine tablets was a concern for recipients.	Implementers discussed challenges in providing consistent funding for cholera response.
Fidelity	Descriptions of CATI delivery varied. Some reported only group sessions, some only door-to-door sessions, and some reported receiving both. The contents of hygiene kits also reportedly varied, and many rural site recipients said they did not receive soap in their kits.	Implementers reported a lack of supply of WASH materials including soap and chlorine tablets.
Feasibility	Feasibility challenges centered around the continued procurement of consumable materials (soap, chlorine tablets, ORS).	Implementers reported many feasibility challenges including supply stockouts and transportation issues in rural mountainous areas.
Penetration	Penetration was not explored at the recipient level.	Implementers reported challenges with reaching all households scheduled for CATIs, and refusal of recipients to accept CATI household visits and materials.
Sustainability	Recipients shared mixed views on the sustainability of CATIs. Few participants expressed the ability to continue handwashing with soap and water treatment behaviors long-term, though they appreciated the intervention. Access and affordability of chlorine tablets and soap were major concerns.	The provision of improved water sources at the community level was a major concern expressed by implementers. Many believed that sustainability depended on improved funding sources and increased cooperation with local leaders.

CATI = case-area targeted intervention.

**Table 6 ijerph-21-00027-t006:** Recommendations for CATI delivery.

Key Component	Recommendations
Handwashing with soap	Develop hygiene promotion using theory-driven and evidence-based behavior change methods. Update hygiene promotion to emphasize that the provided soap is intended for handwashing only. Promoters could note that the longer the soap lasts for handwashing, the longer it will be useful to prevent cholera. Distribute soapy water using water and detergent powder prepared during the session. Soapy water may be a promising alternative to bar soap distribution. Promoters could demonstrate how to make additional soapy water from detergent or small pieces of soap during the group session. Place ash and water at latrines and cooking areas for handwashing with a cleansing agent when soap is unavailable.
Water treatment	Provide clear and consistent instructions on water chlorination protocol. Only deliver one type of chlorine tablet (with the same dosage) in a given village or neighborhood. Include a group session where recipients drink chlorine-treated water together and have a discussion about the taste. Update CATI protocol to clarify the below points: (1) that only drinking water needs to be treated with chlorine, and that bathing water does not require treatment—this will help chlorine tablets last throughout the high-risk period for cholera; (2) the importance that chlorine tablets have in protection from disease despite the strong taste; (3) alternative drinking water treatments such as boiling (heat water until large bubbles form) should be used when chlorine tablets are not available; (4) locations where chlorine tablets can be purchased, and their cost.
ORS	Update CATI protocol to clarify the following points: (1) ORS does not remove germs from water and does not prevent cholera, and (2) ORS is used to treat dehydration.
CATI Delivery	Develop and implement CATIs with the engagement of local stakeholders, such as community chiefs, to support intervention implementation and rapid response, and improve acceptability. Standardize CATI protocols. Ensure complete delivery of all WASH materials, including soap and chlorine tablets. Deliver all CATIs as group sessions with door-to-door visits for further reinforcement when needed. Ensure stocks of WASH materials for CATI delivery before cholera outbreaks.
Vaccination	Discuss any ongoing cholera vaccine campaigns in the areas where CATIs are being conducted and provide information on what the cholera vaccine is and the protection it provides. Explain that the cholera vaccine reduces the risk of cholera but does not eliminate the risk of cholera, which is why preventative WASH behaviors remain important.

CATI = case-area targeted intervention.

## Data Availability

Anonymized data may be made available upon request. Requests should be directed to Christine Marie George (cmgeorge@jhu.edu).

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
