# Peer review of "A Quantitative and Qualitative Program Evaluation of a Case-Area Targeted Intervention to Reduce Cholera in Eastern Democratic Republic of the Congo"

_ijerph, 2023, doi:10.3390/ijerph21010027_

Round 1

Reviewer 1 Report

Comments and Suggestions for Authors

This is an interesting study within the context of Congo. The study's findings are typical to other locations of the world where cholera poses a serious threat. Strengthening WASH is critical to get ahead of cholera outbreaks. However, a narrative in this study may not be what the authors intend.

First, vaccines are essential to the fight against cholera, but it is not the only thing. Authors have provided a narrative suggesting that cholera vaccines are perhaps a panacea, whereas it is also known that multiple doses of vaccines will be required over the years. This highlights the fact that strengthening WASH education is critical to control cholera. Therefore, the discussion section on the importance of vaccines should be reworded to provide limitations of this prevention method.

Second, the environmental-human pathways of cholera must be included in any CATI. This is because vibrios are environmental pathogens and the shedding rates of infectious individuals immensely impact water systems. Authors have chosen not to include such literature evidence while designing their study, for reasons known to the research team. This is questionable because nearly half of the Congo's rural population rely on rivers, streams, and ponds for daily water needs. This leads to another critical question: why discussions on the predictive abilities of cholera are not mentioned in the section? WHO-GTFCC has recently highlighted climate and weather processes' role in cholera outbreaks in Africa.  Colwell and her team have researched the possibilities of predicting cholera to aid decision-making on where and how OCVs, vaccines etc need to be prioritized. It is rather questionable to overlook this piece of evidence and literature and conclude an intervention study.

Comments on the Quality of English Language

No comments

Reviewer 2 Report

Comments and Suggestions for Authors

Abstract:

Line 25: The word DRC should be written in full since its the first time.

Line 34: sample size for qualitative data being mentioned here should come before presenting the findings. Likely it should come immediately after presenting sample size for quantitative data.

Line 37: revise, it should read ....and use of chlorine for water treatment....

Introduction:

Line 77: replace "increasing protective" with "promoting"

Line 80: Commas should be included in between the reference.

Line 90 - 92: Improve English in these sentences

Line 99: replace "on increasing protective with "in promoting"

Methods:

Preferebly, ethical approval section should as a last section of your materials and methods i.e. before the study findings section

Line 125: improve sentence construction.

Line 133 - 134: should read like "The hygiene kit varies between settings depending on the availability of CATI materials."

Line 143: replace "on food hygiene" with "through food hygiene".

Line 152: make it clear, was control area only in Katana and not Bukavu?

Line 155 - 156: It should be made clear that 6 CATI and 6 Controls were obtained from each study location.

Selection criteria of control HH in Katana and Bukavu was different. It could be good if you include a statement in your methods where you had this difference.

Line 167: When you say nearest HH, you meant did you mean HH next to each other? make it clear.

Line 173: Make it clear why the researcher met the village chief only in Katana and not in Bukavu.

The process of selecting CATI and control areas can be clear if you also use a CONSORT flow diagram. Currently it is not straight forward to understand

Line 178 - 179: Any reason why selecting 5 HH?

Line 179: "5" should be in words

Line 184: cleansing agents should specified what it included

Line 199: "availalability of the HH" you mean availability of the household members?

Line 200: Can you include an explanation why observations were conducted in the morning and not afternoon

Line 207 - 211: Can you also define what you mean by food related event.

Line 218: So can you conclude by stating the sample size of your study. Also you should include how qualitative data sample size was determined.

Line 220: Include the number of enumerators and what they were trained on. Also if these are the same enumerators who conducted the observations, then such information should be included accordingly.

Line 231: So what was the type of sampling used?

Line 252 - 253: Thats a CONSORT flow diagram is required in the methods

Line 260: add "on" between "surveyed" and "an"

Line 293: "Very" can be removed

Line 297: should read "water only"

Line 322 - 323: Any explanation why the respondents were able to recall these two handwashing critical moments.

Line 343: With the mentioning of ash, there is a need to include what kind of messages were the study participants receiving. This could be included in the methods section.

Line 348 - 356: What recommendation are you giving on the issue of forgetting to wash hands at critical times?

Line 361: could have been better if we know the % of HH accessing these different water sources.

Line 428 - 436: Also what is your recommendation on the issues that others do not like chlorinated water due to smell - maybe a point for discussion.

Line 493 - 499: There is a need to recommend that CATI interventions should prepare communities to continuously supply materials such as soap on their own. For this to happen, interventions should address what contextual and psychoscial issues are needed for the HHs to sustain availability of soap e.g. for handwashing.

Line 503: For those who had received cholera vaccine, what was the source of information about the vaccine.

Comments on the Quality of English Language

English quality is good, just require minor editions.
